# Mechanisms for Bile Acids CDCA- and DCA-Stimulated Hepatic Spexin Expression

**DOI:** 10.3390/cells11142159

**Published:** 2022-07-10

**Authors:** Qi Lai, Yanhua Ma, Jin Bai, Min Zhuang, Shaofei Pei, Ni He, Junlin Yin, Baomin Fan, Zhaoxiang Bian, Guangzhi Zeng, Chengyuan Lin

**Affiliations:** 1Key Laboratory of Chemistry in Ethnic Medicinal Resources, State Ethnic Affairs Commission & Ministry of Education, School of Ethnic Medicine, Yunnan Minzu University, Kunming 650504, China; 10217160111@ymu.edu.cn (Q.L.); mayanhua@hkbu.edu.hk (Y.M.); minzhuang@hkbu.edu.hk (M.Z.); 10219160130@ymu.edu.cn (S.P.); 10219160114@ymu.edu.cn (N.H.); 041691@ymu.edu.cn (J.Y.); fanbm@ynni.edu.cn (B.F.); 2Centre for Chinese Herbal Medicine Drug Development Limited, Hong Kong Baptist University, Hong Kong, China; bzxiang@hkbu.edu.hk; 3Department of Obstetrics and Gynecology, University of California Irvine, Irvine, CA 92697, USA; baij3@hs.uci.edu; 4School of Chinese Medicine, Hong Kong Baptist University, Hong Kong, China

**Keywords:** spexin, bile acid, liver cells, gene transcription

## Abstract

Spexin (SPX) is a novel peptide involved in glucose and lipid metabolism and suppresses hepatic total bile acid levels by inhibiting hepatic cholesterol 7α-hydroxylase 1 expression. As important mediators for glycolysis/gluconeogenesis and lipid metabolism, the effects of bile acids on SPX expression is yet to be understood. By using SMMC7721 and BEL-7402 cell lines, we screened the effects of bile acids and found that chenodeoxycholic acid (CDCA) and deoxycholic acid (DCA) can stimulate SPX gene transcription. Both CDCA and DCA were able to stimulate SPX mRNA expression in the liver but not colon and ileum in mice. In SMMC7721 and BEL-7402 cells, CDCA- and DCA-induced SPX promoter activity was mimicked by bile acid receptor FXR and TGR5 activation and suppressed by FXR and TGR5 silencing. Adenylate cyclase (AC)/cyclic adenosine monophosphate (cAMP) activators significantly increased SPX promoter activity whereas the inhibitors for AC/CAMP/protein kinase A (PKA) and mitogen-activated protein kinases (MAPK) pathway attenuated CDCA- and DCA-induced SPX transcription. Thus, CDCA and DCA stimulate SPX expression at the hepatic level through FXR and TGR5 mediated AC/cAMP/PKA and MAPK cascades.

## 1. Introduction

The neuropeptide spexin (SPX) belongs to the galanin family coupled with galanin receptor 2 (GALR2) and galanin receptor 3 (GALR3) [1]. It was first identified by data mining of human proteome via hidden Markov method [2]. SPX is widely expressed at the tissue level, including liver, pancreas, visceral fat, intestine, and other peripheral tissues [3,4], suggesting that SPX may have a variety of functions. In a recent study, we have shown hippocampal SPX expression is linked to anxiety-related behaviors in mice [5]. In addition, SPX is closely related to the pathogenesis of glucose and lipid metabolism-related diseases, including obesity, diabetes, and non-alcoholic fatty liver [6]. Compared with the healthy populations, the circulatory SPX levels are lower in obese children [7], and in patients with metabolic syndrome [8]. Recently, SPX has emerged as a new target for drug design/therapeutic strategy, e.g., for type 2 diabetes mellitus (T2DM) treatment [3], gastrointestinal disease treatment [9], and prediction of insulin resistance in patients with nonalcoholic fatty liver disease [10]. We recently reported that SPX plays a role in bile acids synthesis and metabolism and its intraperitoneal injection can decrease cholesterol 7α-hydroxylase (CYP7A1) levels and total bile acids via GALR2/3 receptors in the mouse and rat liver [11]. To date, it is still unknown how bile acids regulate SPX expression at the hepatic level.

The signaling molecule bile acids can mediate the glycolysis/gluconeogenesis and lipid metabolism throughout the body by activating their receptors in liver, intestine, and other peripheral tissues; thus, their functional expressions are very important for the regulation of metabolic diseases including obesity, T2DM, and non-alcoholic fatty liver [12]. The main products of cholesterol catabolism bile acids are synthesized by CYP7A1 in the hepatic endoplasmic reticulum, and play an important role in liver and intestinal diseases [13]. When the bile acids are secreted into the intestine, approximately 95% of bile acids are reabsorbed back to the liver via the portal vein, and the other 5% are excreted in the feces [14]. Bile acids mainly play a regulatory role in glucose and lipid metabolism by binding to farnesoid X receptor (FXR) and G protein-coupled receptor (TGR5), which is affected by bile acid synthesis, other metabolic pathways and physiological states [15]. Bile acids (12α-hydroxylated bile acids) can act as repressors for insulin action and improve the alteration of intestinal flora in insulin resistance and regulate the pathogenesis of T2DM [16]. Prolonged severe cholestasis may lead to liver fibrosis, cirrhosis, hepatocellular carcinoma or cholangiocarcinoma, and even death [17]. Previous studies have shown that ursodeoxycholic acid (UDCA) is widely used in various intrahepatic cholestatic diseases with proven efficacy [18]. In addition, bile acids are also able to regulate levels of obesity-related hormones, i.e., insulin [19] and glucagon-like peptide 1 (GLP-1) [20]. Increasing studies have discovered a variety of functions of bile acids and developed their respective medicinal uses, though the functions of bile acids on obesity-related hormones are yet to be understood.

In this study, we hypothesized that bile acids stimulate SPX gene expression by activating TGR5 and FXR in mouse liver. By using the SMMC7721 and BEL-7402 cells, we showed that chenodeoxycholic acid (CDCA) and deoxycholic acid (DCA) stimulate hepatic SPX gene expression by activating TGR5 and FXR receptors through adenylate cyclase (AC)/cyclic adenosine monophosphate (cAMP)/protein kinase A (PKA) and mitogen-activated protein kinases (MAPK) cascades. To date, our study is the first to report the mechanisms by which bile acids regulate SPX gene expression in vertebrates.

## 2. Materials and Methods

### 2.1. Animals

Male C57BL/6 mice (aged 6–8 weeks) were obtained from the Hunan Slek Jingda Experimental Animal Co., Ltd. (Changsha, China) and maintained in specific pathogen free (SPF) class animal room with constant temperature (22–23 °C) under a 12 h light/12 h dark photoperiod and given ad libitum access to food and water. After 7 days of adaptive feeding, mice were randomly divided into 7 groups (no significant weight difference, *p* > 0.5). CDCA and DCA (Sigma, St. Louis, MO, USA) were dissolved in saline and diluted to 0, 50, 100, and 200 mg/kg for oral gavage as previously described [21]. After the treatment, all of the animals were finally injected with 1% sodium pentobarbital at 50 mg per kg body weight and sacrificed by carbon dioxide inhalation to obtain the tissue samples of liver, colon, and ileum. The tissues were stored in an RNA sample preservation solution, RNA later (TaKaRa Bio Inc., Kusatsu, Japan) at 4 °C. All procedures in this study met the requirements of the animal experimental committee and the ethics committee of Yunnan Minzu University.

### 2.2. Test Substances

DMEM, RPMI-1640, and fetal bovine serum (FBS) were acquired from Biological Industries (Beit Haemek, Israel). Antibiotic-antimycotic, OPTI-MEM, and lipofectamine^TM^ were acquired from Invitrogen (Carlsbad, CA, USA). Pharmacological agents for signaling targets/kinases, including forskolin, CPT-cAMP, MDL12330A, H89, PD169316, U0126, and SP600125 were acquired from Calbiochem (San Diego, CA, USA), and TGR5 receptor agonist (4-Isoxazolecarboxamide, CAS NO: 1197300-24-5) was purchased from Tocris (Ellisville, MO, USA). These test substances were dissolved in double steamed water or dimethyl sulfoxide, stored at −80 °C, and diluted with pre-warmed culture medium to appropriate concentrations 10 min before drug treatment. SiRNA reagents were purchased from Ambion (Shanghai, China). All cell plates were purchased from NEST (Wuxi, China). Other chemicals, unless specified, were purchased from Sigma (St. Louis, MO, USA).

### 2.3. Measurement of SPX Promoter Activity in SMMC7721 and BEL-7402 Cells

To examine the role of promoter activation in bile-acid-induced SPX gene transcription, a 1.2 kb 5′ promoter of human SPX gene was subcloned into the luciferase-expressing reporter pGL3.Basic (Promega, Madison, WI, USA) to generate the pGL3.SPX.Luc construct for transfection studies in SMMC7721 and BEL-7402 cells. For promoter activity expression, SMMC7721 and BEL-7402 cells were maintained in RPMI-1640 containing 10% FBS and DMEM containing 10% FBS at the seeding density of 5 × 10^4^ cells/well in 24 wells, respectively. Briefly, 200 ng pGL3.SPX.Luc, 30 ng TK-Renilla control constructs and 50–200 ng pcDNA3.1 NR1H4 receptor expression vector (Accession No: NM_001206979) were mixed with 1 μL lipofectamine 2000 (Invitrogen, Carlsbad, CA) and applied to ~70% confluent SMMC7721 and BEL-7402 cells (~10^5^ cells/well) in a 24-well plate overnight at 37 °C. The siRNA was also co-transfected with TK-Renilla control constructs and pcDNA3.1 NR1H4 receptor expression vector into the SMMC7721 and BEL-7402 cells. The cells were then treated with the bile acids for the doses and duration in triplicates for 24 h as indicated [22]. After drug treatment, SMMC7721 and BEL-7402 cells were dissolved in passive lysis buffer (Promega) and the cell lysate prepared was subjected to luciferase activity measurement using a Dual-Glo^®^ Luciferase Assay Kit (Promega, Madison, WI, USA). The raw data for firefly luciferase activities (in ALU) was normalized as a ratio of renilla luciferase data detected in the same sample (referred to as the “Relative luciferase activity ratio, RLU”).

### 2.4. Measurement of SPX mRNA Expression in Mice by Real-Time PCR

Total mRNA from colon, ileum, and liver were extracted and reverse transcribed by using PrimeScript™ RT reagent Kit with gDNA Eraser and TB Green^®^ Premix Ex Taq™ II (TaKara) as described previously [5]. The real-time qPCR system was conducted in a 20 µL system under the conditions listed in the Table 1. Comparative CT method (2^−∆∆CT^ method) was used to calculate SPX (Accession No: NM_001369015.1) mRNA expression with 18S rRNA as the internal reference control. SPX mRNA was measured by using the primers (forward: 5′-CTGGTGCTGTCTGCGCTG-3′, reverse: 5′-CTGGGTT TCGTCTTTCTGG-3′), while 18S rRNA was measured as an internal control by using the primers (forward: 5′-GCAATTATTCCCCATGAACG-3′, reverse: 5′-AGGGCCTCACTAAACCATCC-3′) [23].

### 2.5. Western Blot Analysis

SMMC7721 cells were prepared by trypsin/DNase II digestion and incubated at seeding density of 2.5 × 10^6^ cells/well in 6-well plates for 24 h with or without drug treatment. After drug treatment, cells were lysed with 30 μL RIPA regent (Sangon, Shanghai, China), with the presence of 1% thioglycol. The samples were boiled at 100 °C for 5 min and then stored at −20 °C. A total of 30 μg of each sample was separated by sodium dodecyl sulphate-polyacrylamide gel electrophoresis and transferred to polyvinylidene difluoride membranes (Immobilon^®^, Merck Millipore, Cork, Ireland). Subsequently, the membranes were blocked with 5% non-fat milk for 1 h at room temperature and incubated with the primary antibody phosphorylated form (p-) and total protein of the extracellular signal-regulated kinase 1 and 2 (ERK1/2, 1:3000, Cell Signaling Technology, Beverly, MA, USA), p38 (1:3000, Cell Signaling Technology, Beverly, MA, USA), Jun-N-terminal kinase (JNK, 1:3000, Cell Signaling Technology, Beverly, MA, USA), HRP-conjugated β-actin (Proteintech, Wuhan, China) at 4 °C overnight, followed by 1.5 h incubation with horseradish peroxidase-conjugated anti-rabbit secondary antibody (Sangon, Shanghai, China) at room temperature.

### 2.6. Statistical Analysis

Data presented (Mean ± SEM) were pooled from quadruplicates and analyzed with Student’s *t* test or ANOVA followed by Fisher’s Least Significance Difference post hoc. Differences between groups were considered as significant at *p* < 0.05.

## 3. Results

### 3.1. Screening of Bile Acid Monomers with Effects on SPX Promoter Activation

To verify the effect of bile acids on SPX promoter, bile acids, including cholic acid (CA), UDCA, CDCA, DCA, glycodeoxycholic acid (GDCA), taurocholic acid (TCA), and glycocholic acid (GCA), were used to treat SMMC7721 and BEL-7402 cells (Figure 1A). Only CDCA and DCA showed the significantly stimulatory effects on SPX promoter in both SMMC7721 and BEL-7402 cells (*p* < 0.05), while CA, UDCA, GDCA, TCA and GCA did not alter SPX promoter activity in the two cell lines (Figure 1B,E). Treatment with 50 μM CDCA and DCA for 24 h increased mouse SPX promoter activity by 2.54 ± 0.17 folds and 2.94 ± 0.30 folds (*p* < 0.05) in BEL-7402 cells (Figure 1C,D), and by 1.99 ± 0.02 folds and 3.54 ± 0.34 folds in SMMC7721 cells (Figure 1F,G), respectively. By fixing the duration of drug treatment at 24 h, increasing doses of CDCA and DCA (0–100 μM) stimulated SPX promoter activity in a concentration-dependent manner in both SMMC7721 and BEL-7402 cells. CDCA significantly stimulated SPX promoter activity at 50 μM and reached its maximum at 100 μM in BEL-7402 and SMMC7721 cells (Figure 1C,F), while DCA began to stimulate SPX promoter activity at 50 μM, reached its maximum at 100 μM in BEL-7402 cells and only stimulated SPX promoter activity at 100 μM in SMMC7721 cells (Figure 1D,G).

### 3.2. CDCA- and DCA-Induced SPX Expression in Mouse Liver

To assess the effects of CDCA and DCA on SPX expression in C57BL/6 mouse in vivo (*n* = 8), real-time qPCR was used to measure SPX expression levels in colon, ileum, and liver tissues. Compared with the blank control group, CDCA and DCA at 200 mg/kg stimulated SPX mRNA expression levels by 4.39 ± 1.33 folds and 2.37 ± 0.20 folds in mouse liver (*p* < 0.05), respectively (Figure 2A,B). Neither CDCA nor DCA at all concentrations could alter SPX expression levels in the mouse colon or ileum (Figure 2C–F).

### 3.3. Bile Acid Receptor-Mediated CDCA- and DCA-Induced SPX Promoter Activity

TGR5 receptor agonist began to stimulate SPX promoter activity from 24 h and maximized this response at 48 h by 3.22 ± 0.55 folds (*p* < 0.05, Figure 3A). By fixing the duration at 24 h, TGR5 receptor agonist stimulated SPX promoter activity in a dose-dependent manner with the maximal response at 30 µM (Figure 3B). Meanwhile, knocking down TGR5 receptor by transfecting its specific siRNA si-Gpbar1 blocked both CDCA- and DCA-induced SPX promoter activity in SMMC7721 cells (Figure 3C,D). Over-expression of the nuclear receptor FXR construct pcDNA3.1 NR1H4 dose-dependently stimulated SPX promoter activity (Figure 3E), whereas knocking down FXR by transfecting si-NR1H4 also attenuated both CDCA- and DCA-induced SPX promoter activity in SMMC7721 cells (Figure 3F,G).

### 3.4. The Role of AC/cAMP/PKA Pathway in CDCA- and DCA-Induced SPX Promoter Activity

To study the roles of cAMP-dependent cascade in CDCA- and DCA-induced SPX promoter activity, SMMC7721 cells were treated with increasing concentrations of the adenylate cycles (AC) activator Forskolin (10–1000 nM, 24 h) or membrane-permeable cAMP analog CPT-cAMP (0.1–100 μM, 24 h). Forskolin dose-dependently increased SPX promoter activity from 100 nM and reached the maximal level at 1000 nM (2.11 ± 0.12-fold vs. control, *p* < 0.05, Figure 4A). CPT-cAMP was effective to increase SPX promoter activity at 100 nM (1.82 ± 0.04-fold vs. control, *p* < 0.05, Figure 4B). In parallel studies, both CDCA- and DCA-induced SPX promoter activity could be partially attenuated by 2 μM AC inhibitor MDL12330A or 2 μM PKA inhibitor H89 in SMMC7721 cells, respectively (Figure 4C–F).

### 3.5. The Role of MAPK/ERK, MAPK/JNK, and MAPK/p38 Pathway in CDCA- and DCA-Induced SPX Promoter Activity

Given that FXR is known to be functionally coupled with MAPK pathways [24], the potential involvement of the two post-receptor signaling cascades in SPX regulation by CDCA and DCA was examined. In SMMC7721 cells, CDCA-induced SPX promoter activity was totally blocked by 2 μM MAPK/JNK inhibitor SP600125 (Figure 5A) and attenuated by 1 μM MAPK/MEK_1/2_ inhibitor U0126 and 2 μM p38 inhibitor PD169316 (Figure 5C,E), while DCA-induced SPX promoter activity was attenuated by SP600125 (Figure 5B) and totally blocked by U0126 and PD169316 (Figure 5D,F). CDCA and DCA effectively triggered rapid JNK, MEK_1/2_ and P38 phosphorylation in SMMC7721 cells; these responses triggered by CDCA or DCA could be blocked by simultaneous incubation with the JNK inhibitor SP600125, MEK_1/2_ inhibitor U0126 and P38 MAPK inhibitor PD169316, respectively (Figure 5G), suggesting the involvement of MAPK/JNK, MAPK/ERK_1/2_, and MAPK/p38 pathway in CDCA- and DCA-induced SPX expression.

## 4. Discussion

The novel neuropeptide SPX acts as an important factor in regulating feeding behavior [25,26], long-chain fatty acid uptake [27], energy utilization [27,28], and body weight regulation [29,30]. Bile acids also show a mediatory role in lipid metabolism and energy balance [31,32]. We recently reported that SPX inhibits the synthesis of total bile acids (TBA), CA, CDCA, TCA and other bile acids via GALR2/3 receptors to participate in lipid metabolism [11]. However, whether bile acids can regulate SPX is yet to be determined. Bile acids are classified into primary and secondary bile acids by sources, and into free and conjugated bile acids by structures [33]. Primary bile acids process bacterial action in the intestine and undergo 7-alpha dehydroxylation to produce secondary bile acids (DCA and lithocholic acid (LCA)) [34,35]. Free bile acids can combine with glycine or taurine and form the conjugated bile acids (such as TCA, GCA and GDCA, etc.) [36,37]. Herein we screened the regulatory roles of these bile acids in SPX expression by using the SMMC7721 and BEL-7402 cells. The results showed that the free bile acids CDCA and DCA but not others can induce SPX promoter activation in both SMMC7721 and BEL-7402 cell lines. Bile acids are deconjugated, dehydrogenated, and dehydroxylated in the ileum and colon, which leads to the increase in primary bile acids (CA and CDCA) proportions [35]. We then further showed that CDCA and DCA stimulated SPX mRNA expression in the mouse liver but not the colon and ileum, suggesting that the liver is the main target organ of the two double α-hydroxy type free bile acid on SPX expression.

The extensive biological effects of bile acids are mediated through FXR and TGR5 [38]. FXR and TGR5 are widely expressed in various tissues in an overlapping yet distinct pattern. Especially, FXR is highly expressed in the liver, ileum, and colon, while TGR5 is highly expressed in enteroendocrine cells, gallbladder, and bile ducts [39]. Bile acids CDCA and DCA in physiological concentrations are endogenous ligands for FXR, and their bindings can cause the recruitment of downstream co-activators and co-repressors to regulate the expression of genes, mostly related to bile acid synthesis, transport, and metabolism [40]. Meanwhile, bile acids can directly activate the GPCR receptor and are the known endogenous ligands for TGR5 [41]. In our study, we detected TGR5 expression in SMMC7721 and BEL-7402 cells. TGR5 receptor agonist up-regulated SPX promoter activity in a dose- and time-dependent manner. Similarly, over-expression of FXR construct pcDNA3.1NR1H4 also showed a stimulatory effect in SPX promoter activation. Moreover, the specific siRNAs corresponding to FXR and TGR5 significantly inhibited SPX promoter activation induced by CDCA and DCA. These results suggest that CDCA and DCA stimulate SPX expression through FXR and TGR5 receptors. It was demonstrated that bile acids have different binding affinity/activities on TGR5 and FXR. Among the seven bile acids used in this study, CDCA and DCA exert much stronger activation on TGR5 and FXR than the other derivatives [42,43]. Thus, the stimulatory roles of the two double α-hydroxy type BAs in hepatic SPX expression may be due to their greater activation potency of FXR and TGR5 [44].

The activation of the TGR5 receptor leads to the release of Gα subunits and consequentially increases cAMP production for activating PKA and phosphorylating cAMP response element-binding protein (CREB) [45]. Previous studies demonstrated that bile acid (TCDCA) coupled with TGR5 exert anti-inflammatory and immuno-regulatory effects by mediating through the AC-cAMP-PKA signaling pathway [46]. Our study has shown that cAMP analog CPT-cAMP and AC activator forskolin dose-dependently stimulate SPX promoter activity in SMMC7721 cells. CDCA- and DCA-induced SPX promoter activation were blocked by AC inhibitor MDL12330A and PKA inhibitor H89, respectively. Previous studies have shown that bile acids can activate the MAPK pathways to regulate cell proliferation and apoptosis [47]. Inactivation of the JNK pathway can lead to defects in liver regeneration and inhibition of the p38 MAPK pathway can restore the damaged liver regeneration [47]. The phosphorylation of Tyr and Thr to activate ERK plays an important role in the process of liver fibrosis [48,49]. Our results also demonstrated that (1) CDCA and DCA activate ERK_1/2_, JNK and p38 MAPK, and (2) CDCA- and DCA-induced SPX promoter activation are blocked by MEK_1/2_ inhibitor U0126, JNK inhibitor SP600125, and p38 inhibitor, respectively. These results raise the possibility that CDCA- and DCA-induced SPX promoter activation is regulated by TGR5 through the AC-cAMP-PKA pathway and MAPK cascades at hepatocyte level.

How bile acids act and their biological effects depend on their circulating levels all over the body [50,51], which are regulated by a negative feedback mechanism [33,52]. Excessive bile acid levels can lead to cholestatic disease, whereas insufficient bile acid levels can cause hepatic failure [17,53]. Studies involving bile acid signaling regulation of glucose and lipid metabolism suggest that impaired bile acid metabolism may contribute to the pathogenesis of obesity and T2DM [31,54]. Moreover, bile acids can act on the CNS and directly influence the pathologies of neurological and neurodegenerative diseases [55]. Thomas et al. reported that FXR activation in hepatocytes induces the expression of V-Maf Avian Musculoaponeurotic Fibrosarcoma Oncogene Homolog G (MAFG), which in turn inhibits bile acid synthesis and changes the composition of bile acids [56]. Nonetheless, our study showed that CDCA and DCA at high concentration increase SPX mRNA expression in the mouse liver but not the colon and ileum, which is consistent with the results of in vitro experiments. Spexin, the recently identified peptide that is closely related to the pathogenesis of glucose and lipid metabolism-related dis-eases, including obesity, diabetes and non-alcoholic fatty liver diseases, can suppress the bile acid synthesis and play physiological functions on bile acid metabolism [11]. Results in the current study provide evidence that CDCA and DCA at relatively high concentration can stimulate liver SPX expression, which may enhance the negative feedback regulation of SPX on bile acid synthesis, contributing to the balance of bile acid enterohepatic circulation in vivo.

## 5. Conclusions

In summary, our study first identified that CDCA and DCA stimulated SPX expression in the liver in vivo; these stimulations, coupled with TGR5 and FXR, are mediated through the AC/cAMP/PKA, MEK_1/2_/ERK_1/2_, MAPK/JNK, and MAPK/P38 pathways. These findings have enriched the physiological system between bile acids and SPX in the liver, which provide the understandings of the roles of bile acids in obesity and metabolic diseases. However, further investigations are needed to delineate the functional elements in the SPX promoter conferring bile acid induction of SPX gene expression.

## Figures and Tables

**Figure 1 cells-11-02159-f001:**
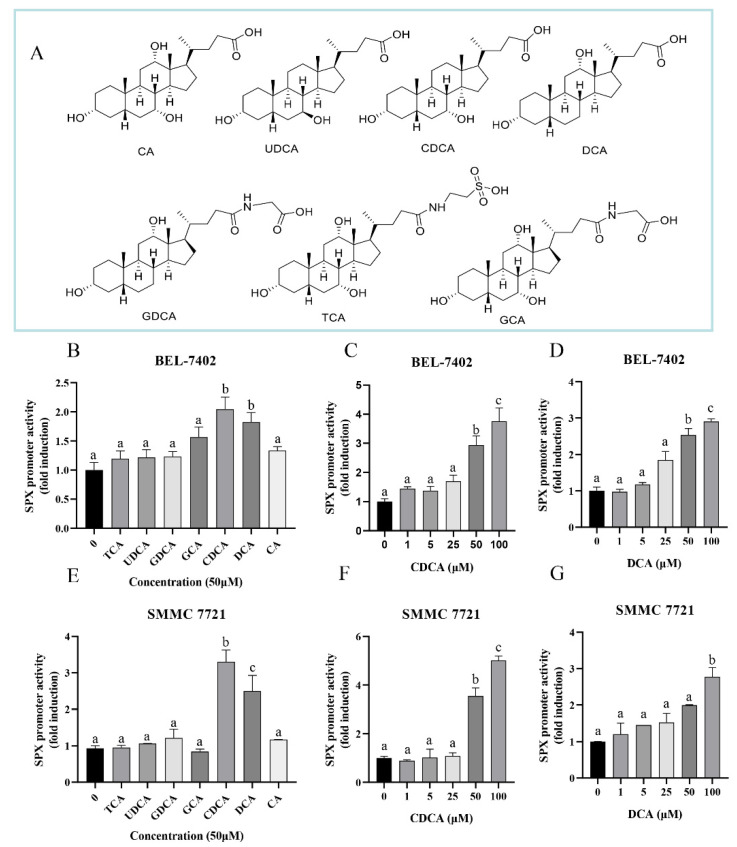
Effects of bile acid on SPX promoter activity were measured in SMMC7721 and BEL-7402 cells. (**A**) Chemical structure of 7 bile acids (TCA, UDCA, GDCA, GCA, CDCA, DCA and CA). (**B**) Effects of different bile acid (50 μM, 24 h) administration on SPX Promoter in BEL-7402 cells. (**C**) Concentration gradient analysis on the effect of CDCA (1–100 μM, 24 h) on SPX promoter activity in BEL-7402 cells. (**D**) Concentration gradient analysis on the effect of DCA (1–100 μM, 24 h) on SPX promoter activity in BEL-7402 cells. (**E**) Effects of different bile acid (50 μM, 24 h) administration on SPX Promoter in SMMC7721 cells. (**F**) Concentration gradient analysis on the effect of CDCA (1–100 μM, 24 h) on SPX promoter activity in SMMC7721. (**G**) Concentration gradient analysis on the effect of DCA (1–100 μM, 24 h) on SPX promoter activity in SMMC7721. The groups denoted by different letters represent a significant difference at *p* < 0.05.

**Figure 2 cells-11-02159-f002:**
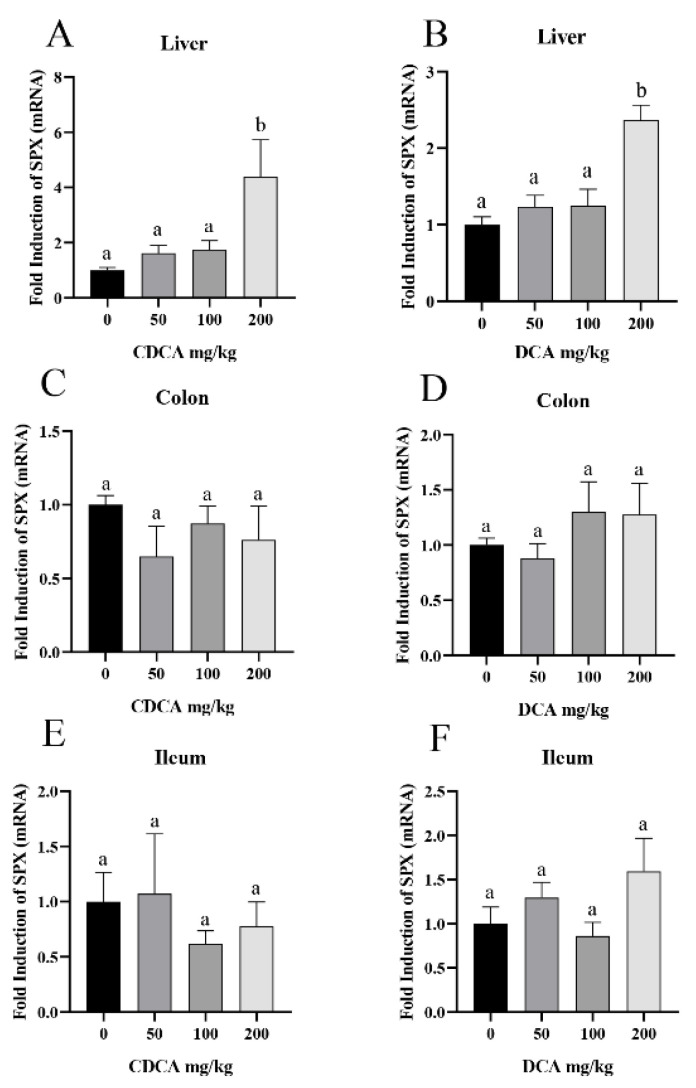
Bile acid promoted the expression of SPX in mouse liver. (**A**) Effect of CDCA on the expression of SPX mRNA in mouse liver. (**B**) Effect of DCA on the expression of SPX mRNA in mouse liver. (**C**) Effect of CDCA on the expression of SPX mRNA in mouse colon. (**D**) Effect of DCA on the expression of SPX mRNA in mouse colon. (**E**) Effect of CDCA on the expression of SPX mRNA in mouse ileum. (**F**) Effect of DCA on the expression of SPX mRNA in mouse ileum. After continuous gavage of bile acid for 7 days, the tissues of mice were collected to measure the expression of SPX mRNA by qPCR. Data were calculated by 2^−∆∆CT^ method, and 0 group was set as unit 1. The groups denoted by different letters represent a significant difference at *p* < 0.05.

**Figure 3 cells-11-02159-f003:**
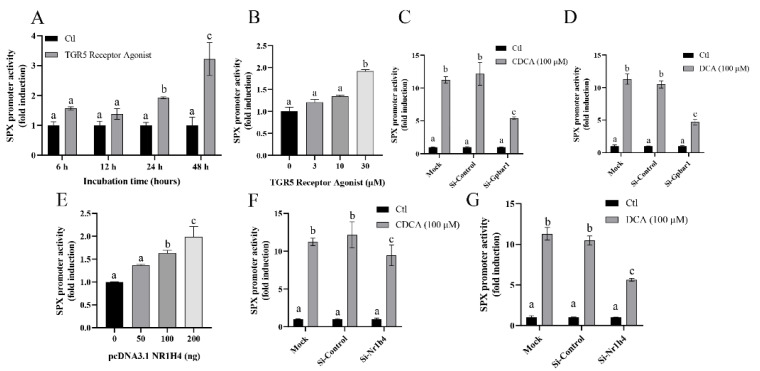
The effect of bile acid on SPX promoter through membrane receptor (TGR5) and nuclear receptor FXR. (**A**) Time course analysis on the effect of TGR5 Receptor Agonist (10 μM, 6–48 h) on SPX promoter activity in SMMC7721 cells. (**B**) Concentration gradient analysis on the effect of TGR5 Receptor Agonist (3–30 μM, 24 h) on SPX promoter activity in SMMC7721 cells. (**C**) SMMC7721 cells transfected with SPX promoter treated by CDCA (100 μM, 24 h) in the presence or absence of siRNA for gpbar1 (30 ng/well) (TGR5) or siRNA control. (**D**) SMMC7721 cells transfected with SPX promoter treated by DCA (100 μM, 24 h) in the presence or absence of siRNA for gpbar1 (30 ng/well) (TGR5) or siRNA control. (**E**) Concentration gradient analysis on the effect of pcDNA3.1 NR1H4 (50–200 ng) on SPX promoter activity in SMMC7721 cells. (**F**) SMMC7721 cells transfected with SPX promoter treated by CDCA (100 μM, 24 h) in the presence or absence of siRNA for NR1H4 (30 ng/well) (FXR) or siRNA control. (**G**) SMMC7721 cells transfected with SPX promoter treated by DCA (100 μM, 24 h) in the presence or absence of siRNA for NR1H4 (30 ng/well) (FXR) or siRNA control. The groups denoted by different letters represent a significant difference at *p* < 0.05.

**Figure 4 cells-11-02159-f004:**
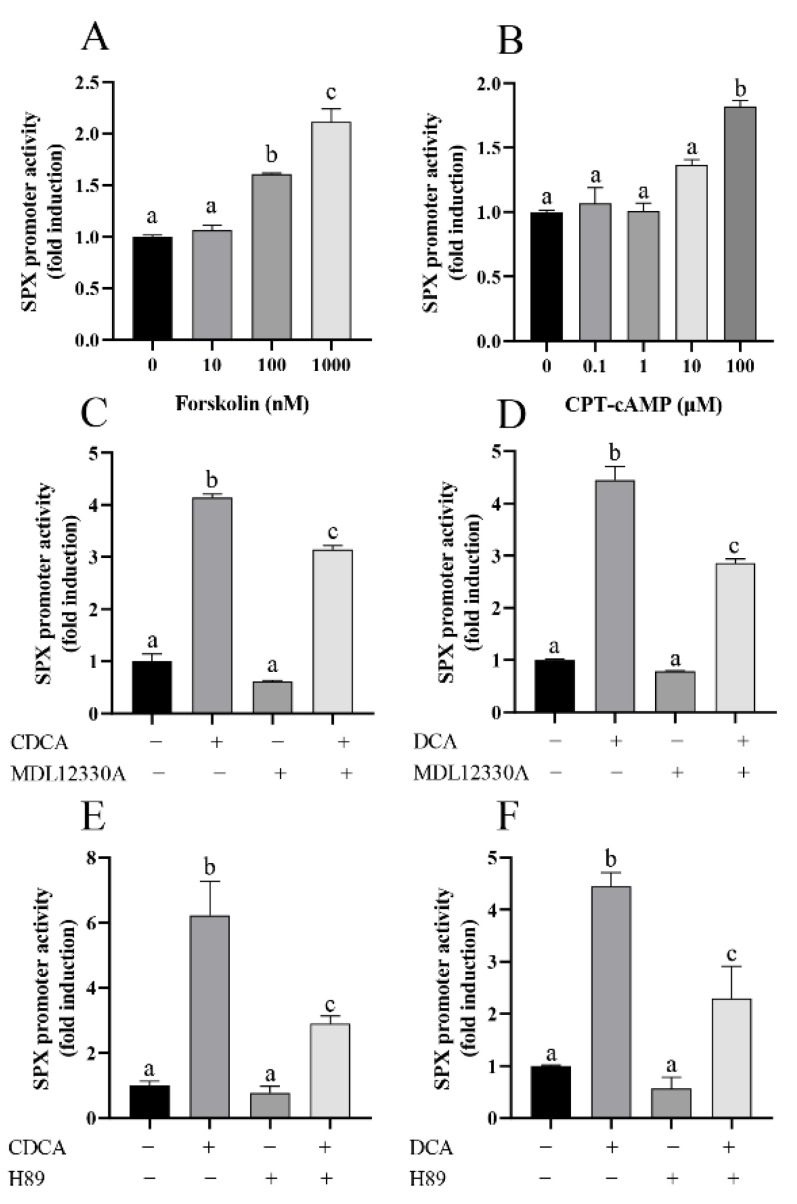
The role of AC/cAMP/PKA pathway in bile acid-promoted SPX promoter activity. (**A**) Dose-dependent (10–1000 nM) effects of forskolin on SPX promoter. (**B**) Dose-dependent (10–100 μM) effects of Cpt-camp on SPX promoter. (**C**) SPX promotor activity was measured with CDCA treatment in the presence or absence of MDL12330A. (**D**) SPX promotor activity was measured with DCA treatment in the presence or absence of MDL12330A. (**E**) SPX promotor activity was measured with CDCA treatment in the presence or absence of H89. (**F**) SPX promotor activity was measured with DCA treatment in the presence or absence of H89. The groups denoted by different letters represent a significant difference at *p* < 0.05.

**Figure 5 cells-11-02159-f005:**
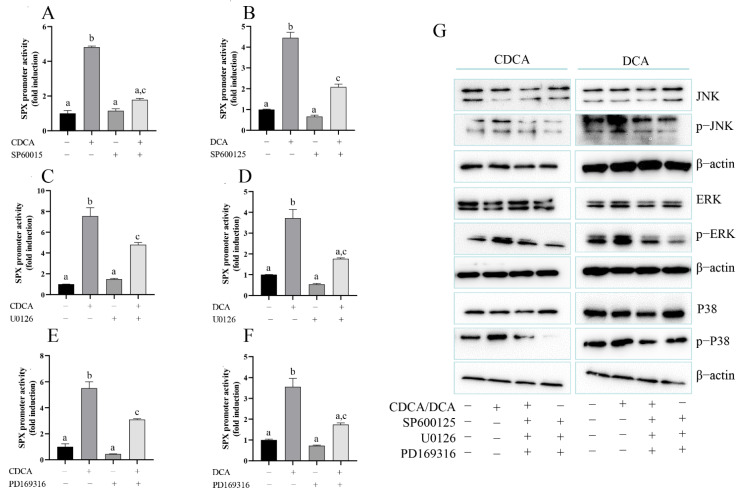
The role of MAPK pathway in bile acid-promoted SPX promoter activity. (**A**) SPX promotor activity was measured with CDCA treatment in the presence or absence of SP600125. (**B**) SPX promotor activity was measured with DCA treatment in the presence or absence of SP600125. (**C**) SPX promotor activity was measured with CDCA treatment in the presence or absence of U0126. (**D**) SPX promotor activity was measured with DCA treatment in the presence or absence of U0126. (**E**) SPX promotor activity was measured with CDCA treatment in the presence or absence of PD169316. (**F**) SPX promotor activity was measured with DCA treatment in the presence or absence of PD169316. (**G**) Effects of CDCA (50 μM, 24 h) and DCA (50 μM, 24 h) on ERK, p-ERK, JNK, p-JNK, P38, and p-P38 were detected by Western blot in SMMC7721 cells. The groups denoted by different letters represent a significant difference at *p* < 0.05.

**Table 1 cells-11-02159-t001:** Detailed information of amplification method for qPCR in mouse liver.

Stage	Temperature (°C)	Time	Cycles
Initial denaturation Denaturation Annealing	95 95 60 60–95	5 min 10 s 30 s	1 35
Melt curve		1.6 °C/s	1

## Data Availability

No new data were created or analyzed in this study. Data sharing is not applicable to this article.

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
