# Peer review of "Mechanisms for Bile Acids CDCA- and DCA-Stimulated Hepatic Spexin Expression"

_cells, 2022, doi:10.3390/cells11142159_

Round 1

Reviewer 1 Report

I have now read the article entitled: Mechanisms for Bile Acids CDCA- and DCA-Stimulated Hepatic Spexin Expression. The work is very interesting and of scientific nature and shows clearly some of the mechanisms involved in the action of these bile acids in lipid metabolism. I have the following questions:

1.       Why Only CDCA and DCA showed the significantly stimulatory effects on SPX promoter in the cell lines?

2.       What are the Implications of these findings on lipid, CHO metabolism, cholestatic disease and Neuro-degenerative diseases?

3.       The doses of CDC and CDCA of 50 and 100 μM seem to be too large compared to the in vivo transient levels of these bile acids. Are the results obtained questionable at this time?

4.       The authors reported that CDCA and DCA were able to stimulate SPX mRNA expression and Promoter activity in cell lines. Have they estimated the real circulating SPX protein levels in the incubates? I have not seen SPX levels in serum following CDC treatment.

5.       What are the effects of circulating levels of SPX on bile acids activity, various roles in metabolism and disease?

6.       It is too early to conclude that the results may shed lights on the roles of bile acids in obesity and metabolic disease since too high concentrations of Bile acids were used.

Reviewer 2 Report

The present manuscript aimed to investigate the role of bile acid, such as chenodeoxycholic acid (CDCA) and deoxycholic acid (DCA) in the neuropeptide spexin (SPX) expression, a novel hormone involved in glucose and lipid metabolism. The Authors found that CDCA and DCA can stimulate SPX hepatic expression both in vivo and in vitro, in particular in SMMC7721 and BEL-7402 cell lines, through bile acid receptor FXR and TGR5 coupled adenylate cyclase (AC)/cyclic adenosine monophosphate (cAMP)/protein kinase A (PKA) pathway and mitogen-activated protein kinases (MAPK) cascades. The paper is very interesting but some points need to be improved:

-      1. In the Abstract, they wrote that only CDCA and DCA were able to stimulate SPX expression. Moreover, in the Results section, they mentioned the use of other bile acids and they show some data in Figure 1, but additional info about these other bile acids are missing in Materials and Methods section.

-       2. Once you spelled out an acronym in the Abstract, it is not more necessary repeat it along the text.

-       3. They have used two hepatocellular cell lines (SMMC7721 and BEL-7402), why? If they want to evaluate the role of bile acids in vitro just in liver, they need to add at least a non-malignant hepatic cell line to better compare the results. Then, if they want to confirm the data in vivo, they need to add cell lines from colon and ileum, control and tumoral, otherwise the data presented are not enough to support the conclusions.

-      4. In Materials and Methods section, the Authors should add references related to the doses of bile acids utilized both in the treatments in vivo and in vitro.

-       5. The Authors performed Western blot analysis in vivo to study the pathway, they should add the same approch for the sampels in vitro to better compare the results.

6. Several syntactic and spelling errors are present, which need to be amended

Round 2

Reviewer 1 Report

I have now read all the responses of the authors to my concerns regarding their manuscript entitled: Mechanisms for Bile Acids CDCA- and DCA-Stimulated 2 Hepatic Spexin Expression.

I think they have  answered the issues properly and I am satisfied that the article should be accepted for publication in Molecules. 

Reviewer 2 Report

The Authors addressed all my points